# The impact of intermittent preventive treatment in school aged children with dihydroartemisinin piperaquine and artesunate amodiaquine on IgG response against six blood stage *Plasmodium falciparum* antigens

Eric Lyimo[1,2,3]*, Geofrey Makenga[1,4], Louise Turner[3,5], Thomas Lavstsen[3,5], John P. A. Lusingu[1], Jean-Pierre Van geertruyden[4], Daniel T. R. Minja[1], Christian W. Wang[3,5], Vito Baraka[1]

1 Tanga Research Centre, National Institute for Medical Research, Tanga, Tanzania, 2 Mwanza Research Centre, National Institute for Medical Research, Mwanza, Tanzania, 3 Department of Immunology and Microbiology, Centre for translational Medicine and Parasitology, University of Copenhagen, Copenhagen, Denmark, 4 Global Health Institute, University of Antwerp, Antwerp, Belgium, 5 Department of Infectious Diseases, Copenhagen University Hospital, Copenhagen, Denmark

⊕ These authors contributed equally to this work.

* ericlyimo@gmail.com

**Data Availability Statement:** The full datasets used and analysed in this study are available upon

## Abstract

Several interventional strategies have been implemented in malaria endemic areas where the burden is high, that include among others, intermittent preventive treatment (IPT), a tactic that blocks transmission and can reduce disease morbidity. However, the implementation IPT strategies raises a genuine concern, intervening the development of naturally acquired immunity to malaria which requires continuous contact with parasite antigens. This study investigated whether dihydroartemisinin-piperaquine (DP) or artesunate-amodiaquine (ASAQ) IPT in schoolchildren (IPTsc) impairs IgG reactivity to six malaria antigens. An IPTsc trial in north-eastern Tanzania administered three doses of DP or ASAQ at four-monthly intervals and the schoolchildren were followed up. This study compared IgG reactivity against GLURP-R2, MSP1, MSP3, and CIDR domains (CIDRa1.1, CIDRa1.4, and CIDRa1.5) of *Plasmodium falciparum* erythrocyte membrane protein 1 (PfEMP-1) in intervention and control groups using enzyme linked immunosorbent assay (ELISA) technique. During the study, 369 schoolchildren were available for analysis, 119, 134 and 116 participants in the control, DP and ASAQ groups, respectively. Breadth of malaria antigen recognition increased significantly during and after the intervention phases and did not differ between the study groups (Trend test: DP, z-score = 5.92, $p < 0.001$, ASAQ, z-score = 6.64, $p < 0.001$ and control, z-score = 5.85, $p < 0.001$). There were no differences between the control and ASAQ group in the recognition of any of the tested antigens at all visits. In the DP group, however, during the intervention period IPTsc did not impair antibody against MSP1, MSP3, CIDRa1.1, CIDRa1.4 and CIDRa1.5, but it did impair against GLURP-R2.

reasonable request from the corresponding author (EL), subject to approval by the Medical Research Coordinating Committee (MRCC) of the National Institute for Medical Research (NIMR) in Tanzania. In accordance with the Ethics Committee's guidelines and institutional data transfer policy, the MRCC mandates that all data collected within Tanzania cannot be transferred or shared without their authorization and the signing of a data transfer agreement. Researchers who meet the criteria for data access may request the datasets by contacting: ethics@nimr.or.tz.

**Funding:** The Flemish Interuniversity Council (VLIRUOS), Belgium (TEAM initiative, grant number TZ2017TEA451A102 awarded to GM), funded the main study, the clinical trial. This study was funded by the Malaria Research and Capacity Building for field trials in Tanzania (MaReCa) project as part of the EDCTP2 programme supported by the European Union (Grant number TMA2015SF–998-MaReCa awarded to JPAL).

**Competing interests:** The authors have declared that no competing interests exist.

The current study has shown that effective IPTsc with DP or ASAQ does not interfere with the development of antibodies against malaria antigens of the blood stages, suggesting that the advancement of naturally acquired immunity to malaria is not impeded by IPTsc interventions.

## Introduction

Malaria is still a major global health problem, with an estimated 249 million cases and 608,000 deaths in 2022, and Tanzania is among the four countries that accounted for over half of all global deaths [1]. The fight against malaria mortality and morbidity has been moderately achieved through several strategies, such as the widespread use of insecticide-treated mosquito nets (ITNs) and indoor residual spraying (IRS), which aim to control the malaria vector. Additional strategies include intermittent preventive treatment (IPT), seasonal malaria chemoprevention (SMC) for transmission intervention, malaria rapid diagnostic tests (mRDTs) used for case detection at the community level and widespread use of artemisinin-based combination therapies (ACTs) for the treatment of *Plasmodium falciparum* infection [2]. In areas with high malaria transmission, the WHO recommends IPT for high-risk groups, infants and pregnant women, and SMC for children, also referred to as intermittent preventive treatment in children (IPTc) which includes mostly under five years old children [2, 3]. However, following the application of efficient measures to control malaria in high-risk groups, including children aged under five years, the malaria burden has broadened to include school age children, above 5 years of age [4, 5]. Schoolchildren have recently captured the attention of researchers after being identified as a group with the highest *P. falciparum* infection rate but with low mortality and morbidity, thus making them a prominent reservoir for transmission [4, 6, 7]. Recent reviews have recommended intermittent preventive treatment in schoolchildren (IPTsc) as the optimal option in terms of acceptability and feasibility that may reduce malaria burden in schoolchildren residing in high transmission areas [6, 8]. Clinical trials have shown that IPTsc with sulphadoxine-pyrimethamine (SP) [9] or ACTs [10] are safe and effective when administered every four months for one year, and the efficacy could last one year after the last dose [10].

The use of IPT though, raises a vital concern that it may hamper the development of naturally acquired immunity to malaria. Immunity to disease develops with continued exposure to malaria antigens and IPT may reduce this exposure [11] and hence when chemoprophylaxis is stopped, the risk of malaria increases [12]. Previous studies have investigated the impact of IPT on antibody development: Intermittent preventive treatment in pregnancy (IPTp) and in infants (IPTi) with SP [13, 14], IPTc with artesunate plus SP [15], and found that IPT reduces antibody development against *P. falciparum*. However, other studies have found no effect on the antibody development after IPTi with SP [14, 16].

In Tanzania, SP has long been barred from use as a first line treatment option because of circulating SP-resistant parasites [17, 18]. Therefore, the use of effective artemisinin and artemisinin-derivatives for IPT in combination with other drugs are appealing. In a study by Makenga et al., [10], it was found that IPTsc with dihydroartemisinin-piperaquine (DP) or artesunate-amodiaquine (ASAQ) delivered in three doses at four-monthly intervals was effective in subsiding *P. falciparum* parasitaemia and reducing clinical malaria and related morbidities, 12 months after the first dose.

In the present study, we investigated the IgG reactivity to six different antigens of *P. falciparum* among schoolchildren in the IPTsc study by Makenga et al [10] which primarily aimed to evaluate changes in mean haemoglobin concentration, malaria incidence, and parasitaemia

prevalence at 12 and 20 months, comparing intervention groups with the control group. In the present study the six antigens were the three cysteine-rich interdomain region (CIDR) domains (CIDRa1.1, CIDRa1.4 and CIDRa1.5) of *P. falciparum* erythrocyte membrane protein 1 (PfEMP1), expressed on the surface, enabling the sequestration of infected erythrocytes to the endothelium of capillaries and venules and associated with severe malaria [19, 20], and three proteins associated with the parasite's merozoite stage: glutamate-rich protein region 2 (GLURP-R2) and merozoite surface proteins 1 (MSP1) and 3 (MSP3). Antibodies against CIDRa1 domains have been shown to be associated with development of immunity against severe malaria and are acquired early in life [21, 22] however, they are lost if transmission is interrupted, indicating that continued exposure is required to maintain the production of the antibodies [22]. Furthermore, antibodies against GLURP-R2, MSP1, MSP3 and other merozoite antigens have been shown to control asymptomatic malaria infections [23]. Antibodies against GLURP have been associated with reduced risk of clinical malaria in Burkinabe and Ghanaian children [24] and associated with the reduction in parasitaemia and febrile episodes of malaria in high and low malaria transmission areas [25, 26].

## Materials and methods

### Study design, site, and participants

The present study is a secondary analysis of data retrieved from a previously conducted randomized, controlled, open-label trial which enrolled 1,566 schoolchildren in Muheza District, Tanga, north-eastern Tanzania (Fig 1) [10], the trial was registered with ClinicalTrials.gov (NCT03640403). The present study analysed samples from 369 schoolchildren who were seen at more than one visit and had samples that passed the quality control (S1 Fig). In brief, the trial assessed the effectiveness and safety of DP and ASAQ as IPTsc in highly malaria-endemic areas. The study had six, visits four months apart and baseline was between 26 March and 30 April 2019, and the last visit was completed on 31 December 2020. The study drugs were administered at baseline (month 0), the second (month 4), and the third visit (month 8). Participants were followed up, and no study drugs were given during the last three visits (months 12, 16, and 20). The present study retrieved and accessed the deidentified data from the main trial in October 2022.

The children were recruited from seven primary schools located in seven different villages in Muheza District, Tanga, north-eastern Tanzania. Schools were selected as previously described [27], in brief, schools in the Muheza District villages with the highest malaria prevalence based on malariometric surveys [28] were chosen. The study included schoolchildren between 5 and 15 years of age with no signs of malaria or clinical features of anaemia. Each of the seven villages namely: Bwitini, Heinkele, Kwakibuyu, Mhamba, Mkulumilo, Pangamlima, and Songa Kibaoni, had a single primary school. Children in class five or lower from each school were enrolled and randomly assigned to the three study arms, control, DP and ASAQ groups. Children aged 11 or older provided informed assent, whereas informed consent from parents or guardians was obtained from children less than 11 years. The study area has two distinct periods of peak malaria transmission, the first occurs after the heavy rains from March to May, while the second follows the moderate rains and is from December to January. The study period from baseline to visit six was 20 months [10, 27]; the first visit was conducted between March and April 2019, during which the study area experienced a long rainy season and before malaria transmission reached its highest rate. The study drugs were given at visit one (month 0), visit two (month 4) and visit three (month 8) after enrolment. Visit two was conducted in August 2019 when malaria transmission was at its first highest annual peak (spanning between July and August) and visit three was conducted in January 2020 after a short rainy season

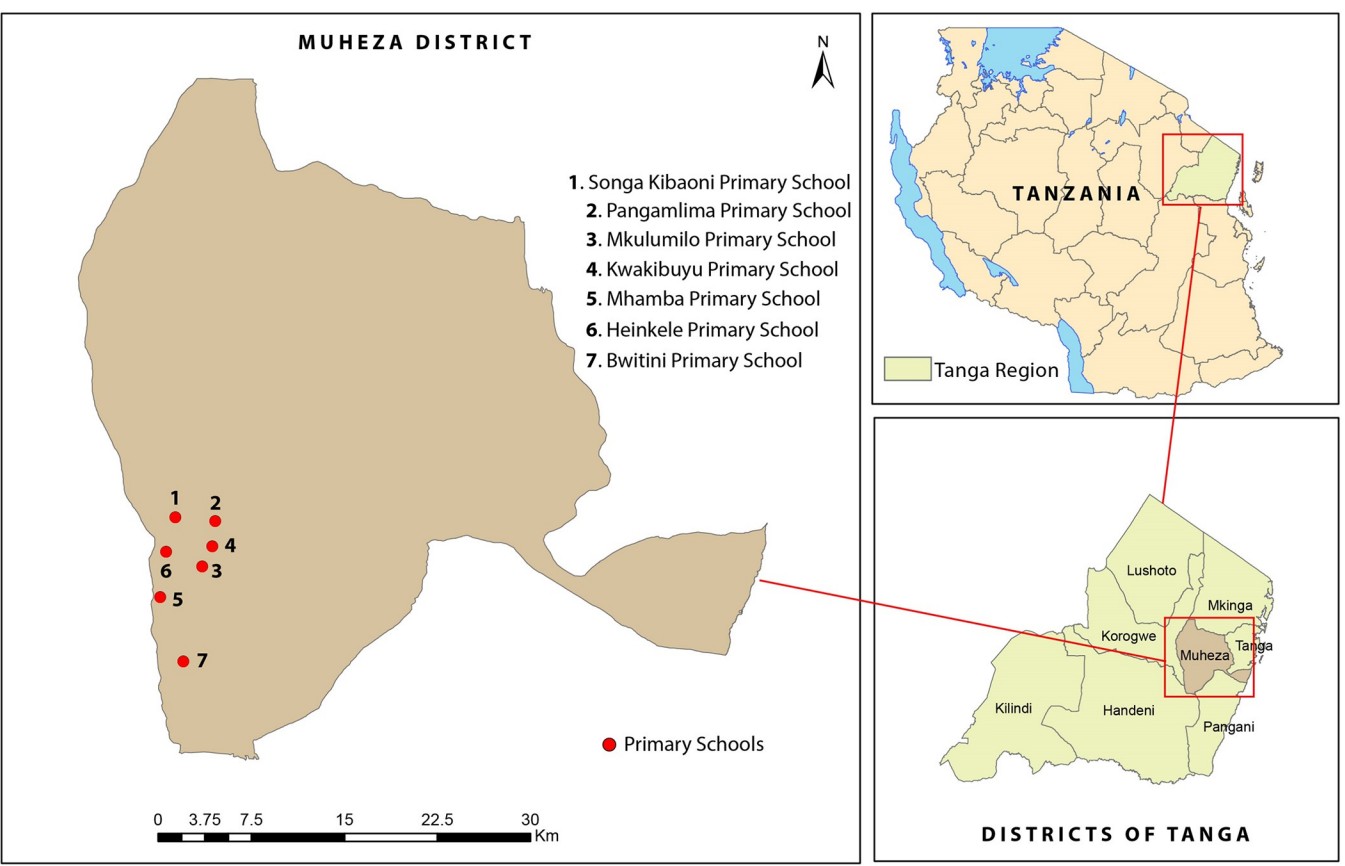

**Fig 1. A map of the study site in north-eastern Tanzania showing the Muheza District and Tanga region.** The figure was generated using QGIS version 3.34.12, based on geospatial data from GPS coordinates of the study sites (the primary schools).

when transmission was in its second annual peak (spanning between December and January) [28]. After the first year of three intervention visits, the schoolchildren were visited three times in the second year, visit four (month 12), visit five (month 16) and visit six (month 20) during which no study drugs were administered. During the fourth visit which was conducted in March 2020, and due to an outbreak of the Severe Acute Respiratory Syndrome-2 (SARS-CoV-2) and the WHO declaring the Coronavirus disease (COVID-19) a pandemic, only two schools; Pangamlima and Songa Kibaoni schools were visited by the study team before the closure of public gatherings and activities.

In the previous trial study [10], the administration of DP and ASAQ was carried out in a carefully monitored process, with dosing tailored to each child's bodyweight to ensure effective treatment for malaria, while also addressing the potential impact of coexisting parasitic infections such as soil-transmitted helminths and schistosomiasis. The study drugs were administered orally following a full therapeutic course over three days based on the child's bodyweight and in accordance with the manufacturer's guidelines. Schoolteachers were trained to administer the medications under the supervision of study nurses.

The dihydroartemisinin (40 mg) and piperaquine (320 mg) tablets (D-Artepp, produced by Guilin Pharmaceutical, Shanghai, China) were donated to the trial study. The dosing regimen for DP was as follows: one tablet for children weighing 11 to < 17 kg, one and a half tablets for those weighing 17 to < 25 kg, two tablets for those weighing 25 to < 36 kg, three tablets for those weighing 36 to < 60 kg, and four tablets for those weighing 60 to < 80 kg. The ASAQ

was supplied by Sanofi Pharmaceuticals (Winthrop), with doses adjusted for bodyweight: one tablet (50 mg artesunate, 135 mg amodiaquine) for children weighing 12 to < 18 kg, one tablet (100 mg artesunate, 270 mg amodiaquine) for children weighing 18 to <36 kg, and two tablets of the higher dose for children weighing ≥ 36 kg.

The DP and ASAQ regimens used in this study have been certified by the WHO as meeting international standards for quality, safety, and effectiveness. Additionally, to address potential impacts of endemic soil-transmitted helminths and schistosomiasis on anaemia in the study area, all participants were treated with albendazole (400 mg) administered orally at the study's start and again one year later during routine mass drug administration by the Tanzanian Neglected Tropical Diseases Control Program. Additionally, children diagnosed with schisto-somiasis received oral praziquantel at a dose of 40 mg/kg.

## Sample collection and processing

The schoolchildren provided finger prick blood samples at baseline, and in all other visits, dried blood spot (DBS) samples were made on Whatman 3 MM paper and haemoglobin concentration was measured with HemoCue® haemoglobin analyser (HemoCue AB, Angelholm, Sweden). As previously described [10], anaemia was defined based on the HemoCue® results and WHO age-specific haemoglobin cutoff points: <11.5 g/dL for children aged 6–11 years, <12.0 g/dL for those aged 12–14 years, <13.0 g/dL for boys aged 15 years, and <12.0 g/dL for girls aged 15 years. The DBS samples were air dried overnight, stored in a Ziplock bag with desiccants, and then stored at -20˚C until testing. At baseline, before treatment, thick and thin blood smears were made to determine parasitaemia. A total of 1,699 DBS samples from 369 schoolchildren were available for the current study. Parasitaemia was calculated as parasite density by using the formula: parasite density/µl = (Number of parasites counted × 8,000) / Number of leukocytes counted, based on an assumed white blood cell count of 8,000/µl [27].

## GLURP-R2, MSP1, MSP3 and PfEMP1 IgG ELISA

The sandwich enzyme-linked immunosorbent assay (ELISA) method was used to assess the antibody recognition of GLURP-R2, MSP1, MSP3 and three PfEMP1 domains (CIDRa1.1, CIDRa1.4 and CIDRa1.5) as previously described [29, 30], with slight modifications. To elute IgG from DBS, a 6.35 mm diameter hole puncher (Staples 10573-CC) was used to cut two DBS discs into a 96-deep-well plate which was then incubated with phosphate buffered saline (PBS) overnight at 4˚C. The quality of elute was assessed as previously described [31]; DBS discs that remained reddish-brown with pale PBS indicated incomplete elution and such samples were not included in the analysis. The DBS eluates were diluted 1:100 in PBS following which the ELISA assay was carried out in a 96-well microtiter plates (Nunc MaxiSorp). Initially, 5 µg/mL of each antigen were used to coat the microtiter plates overnight at 4˚C, excess antigens were discarded. Plates were blocked with 3% skimmed milk in PBS for one hour, then plates were washed three times, and 50 µL of each sample were then added to each well. Rabbit anti-human IgG HRP conjugated antibody (Dako, P0214) was added (1:3000) and incubated for 1 hour at room temperature to identify bound human IgG following which the microtiter plates were washed three times. Afterwards, 3,3',5,5'-tetramethylbenzidine (TMB, T5525, Sigma-Aldrich) was added to detect bound human IgG. Optical densities (ODs) were measured using a Multiskan FC microtiter plate reader (Thermo Scientific) at 450 nm. The DBS samples from naïve Danish blood donors were used as negative controls in the analysis. To ensure that the donors had no prior exposure to the malaria parasite, they were specifically asked if they had not travelled to malaria-endemic countries. The values greater than three standard deviations from the mean of the naïve samples were used as the cut-off. Each participant's breadth of

malaria antigen recognition was scored from 0 to 6 based on their IgG reactivity to the six antigens, as previously described [32]. In brief, the breadth of malaria antigen recognition refers to the range of the studied malaria antigen; GLURP-R2, MSP1, MSP3, and three PfEMP1 domains (CIDRa1.1, CIDRa1.4, and CIDRa1.5), for which a participant's immune system tests positive. Each participant was assigned a score ranging from 0 to 6, reflecting the number of these six specific malaria antigens their immune system is positive for. A score of 0 indicates that the participant's immune system was not positive for any of the six antigens, while a score of 6 indicates that their immune system was positive for all six antigens.

## Statistical analysis

Data from the baseline survey and follow-up visits were managed as previously described [10, 27] Further data were analysed using STATA version 15.0, Microsoft Excel, and R 4.3.1 statistical softwares. Antibody responses to the six *P. falciparum* antigens were analysed as ELISA optical densities (ODs), first, while the breadth of malaria antigen recognition was analysed as a binary response to the six antigens. Trend test across ordered groups was used to assess the trend in serorecognition across the study visits. Secondly, the antibody levels were analysed as continuous variables for each of the six targets; data were set as panel data in STATA and longitudinally analysed using a linear mixed model. Additionally, haemoglobin concentrations were also analysed using the linear mixed model which compared the changes within an individual subject and between study groups. The schoolchildren cohorts as observed in this study, usually have high rates of loss to follow-up, missing data and visits but the linear mixed model is robust enough to account for the missing data [33]. For the comparisons of malaria and anaemia between the treatment groups, proportions were compared. Analyses were also done on only schoolchildren who attended all visits, however, excluding visit four for reasons stated earlier. A *p*-value less than 0.05 was considered statistically significant.

## Ethical approval

The study received ethical approval from the Medical Research Coordinating Committee of Tanzania (Reference number; NIMR/HQ/R.8a/Vol.IX/4120) for this study and (NIMR/HQ/R.8a/Vol.IX/2818, NIMR/HQ/R.8c/Vol.I/668 for amendment, and NIMR/HQ/ R.8c/Vol.I/1276) for renewal of the main study. Also, the study received authorization from the local government authorities and local school committees. Parents or guardians provided written informed consent for their children to participate, while children aged 11 or older were requested to assent. The main study also received regulatory approval from the Tanzania Medicines and Medical Devices Authority (Reference number TFDA0017/CTR/0018/07).

## Results

### Study participants

Blood samples from 369 schoolchildren from a previous trial [10] were available for the current study. At baseline, there were 119 participants in the control group, 134 in the DP group and 116 in the ASAQ group (Table 1). The three groups were similarly matched by sex, anaemia, bed net use, haemoglobin levels, and malaria prevalence (Table 1).

Samples from participants who met the per-protocol criteria in the previous study were used in the present study. Furthermore, during the laboratory analysis, DBS samples that did not pass quality control [31] were removed in the final analysis. In the previous study, participants were excluded from the per-protocol analysis if they had received incomplete treatment,

**Table 1. Participants' characteristics at baseline pre-IPTsc.**

| Characteristic | Control (n = 119) | DP* (n = 134) | ASAQ* (n = 116) | p value |
|---|---|---|---|---|
| **Sex** | | | | |
| Male, n (%) | 62 (52.1) | 69 (51.5) | 64 (55.2) | 0.83 |
| Female, n (%) | 57 (47.9) | 65 (48.5) | 52 (44.8) | |
| **Age** | | | | |
| median [IQR] | 10 [7–11] | 9 [8–11] | 10 [8–12] | 0.44 |
| **Age groups** | | | | |
| 5-9yrs, n (%) | 58 (48.7) | 73 (54.5) | 53 (45.7) | 0.37 |
| 10-15yrs, n (%) | 61 (51.3) | 61 (45.5) | 63 (54.3) | |
| **Malaria prevalence by microscopy** | | | | |
| Negative n (%) | 86 (75.4) | 89 (69.5) | 77 (71.3) | 0.58 |
| Positive n (%) | 28 (24.6) | 39 (30.5) | 31 (28.7) | |
| **Episodes of malaria in the last month** | | | | |
| Negative, n (%) | 68 (57.1) | 81 (60.4) | 61 (52.6) | 0.46 |
| Positive, n (%) | 51 (42.9) | 53 (39.6) | 55 (47.4) | |
| **Bed net use** | | | | |
| No, n (%) | 18 (15.1) | 32 (23.9) | 27 (23.3) | 0.17 |
| Yes, n (%) | 101 (84.9) | 102 (76.1) | 89 (76.7) | |
| **Haemoglobin median** | | | | |
| Mean [SD] | 11.8 [1.5] | 11.6 [1.2] | 11.5 [1.3] | 0.38 |
| **Anaemia** | | | | |
| Anaemia, n (%) | 54 (45.4) | 58 (43.3) | 66 (56.9) | 0.08 |
| Normal, n (%) | 65 (54.6) | 76 (56.7) | 50 (43.1) | |
| **Primary Schoolchildren** | | | | |
| Pangamlima, n (%) | 16 (13.8) | 23 (17.2) | 14 (11.8) | 0.99 |
| Songa Kibaoni, n (%) | 20 (17.2) | 19 (14.2) | 22 (18.5) | |
| Heinkele, n (%) | 16 (13.8) | 24 (17.9) | 17 (14.3) | |
| Kwakibuyu, n (%) | 16 (13.8) | 18 (13.4) | 17 (14.3) | |
| Mhamba, n (%) | 18 (15.5) | 18 (13.4) | 16 (13.4) | |
| Bwitini, n (%) | 13 (11.2) | 14 (10.4) | 13 (10.9) | |
| Mkulumilo, n (%) | 17 (14.7) | 18 (13.4) | 20 (16.8) | |

*Dihydroartemisinin-Piperaquine (DP), Artesunate-Amodiaquine (ASAQ)

missed assessment visits, discontinued treatment, or were treated for anaemia or clinical malaria [10].

## Breadth of malarial antigens recognition during and after IPTsc

The breadth of the six malarial antigens recognition was compared between schoolchildren in the three study groups; DP, ASAQ, and control, at baseline and across all study visits. The number of malaria antigens recognised significantly increased during the study period in all three groups and there was no significant difference in the trend of the breath of malaria antigen recognition between the study groups (Trend test across ordered groups: DP, z-score = 5.92, $p < 0.001$, ASAQ, z-score = 6.64, $p < 0.001$ and control, z-score = 5.85, $p < 0.001$) during and after the intervention phase (Fig 2). Compared with the control group, from baseline the DP group had non-significantly lower difference in mean change in breadth of malaria antigen recognition in all study visits (Table 2) while the ASAQ group had non-

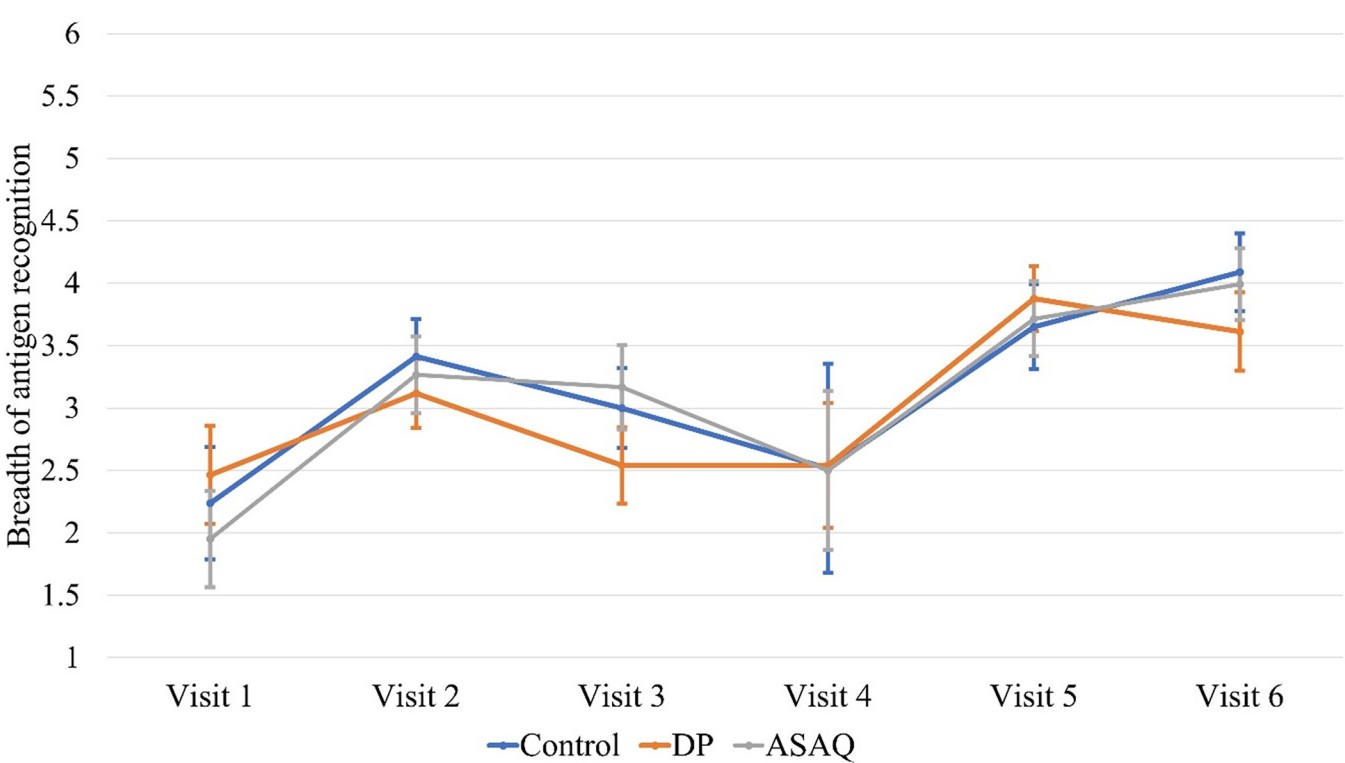

**Fig 2. Changes in the breadth of malaria antigen recognition (means with 95%CI) across six study visits four months apart.** In year-one (visits 1 to 3) DP and ASAQ were given to schoolchildren, and in year-two (visits 4 to 6) no study drugs were given.

significantly higher difference in mean change in breadth of malaria antigen recognition (Table 3).

### IgG reactivity to GLURP-R2, MSP1, and MSP3

The IgG reactivity was analysed longitudinally in the DP and ASAQ groups. The changes in mean IgG responses against GLURP-R2, MSP1, and MSP3 in the ASAQ group were not significantly different from those in the control group across all study visits (Fig 3A–3C and S1 Table). In the DP group, however, IgG reactivity to GLURP-R2 was significantly lower at all follow-up study visits (Fig 3A and S1 Table) and lower against MSP3 at visit six (mean OD -0.06 [95% CI -0.12 –-0.00] $p$ = 0.039) (S1 Table and Fig 3C). The IgG reactivity to MSP1 in the DP group was not significantly different from the control group in either analysis (Fig 3B and S1 Table).

**Table 2. The impact of IPT with DP on the breadth of malaria antigen recognition.**

| Study visit (Month) | N | Mean (Control) | N | Mean (DP) | Difference in mean change in breadth of malaria antigen recognition from baseline (95% CI) | p value |
|---|---|---|---|---|---|---|
| 1 (0) (Baseline) | 60 | 2.27 | 71 | 2.47 | – | |
| 2 (4) | 115 | 3.42 | 130 | 3.12 | -0.50 (-1.22–0.22) | 0.18 |
| 3 (8) | 107 | 3.00 | 126 | 2.52 | -0.68 (-1.41–0.05) | 0.07 |
| 4 (12) | 29 | 2.52 | 39 | 2.51 | -0.20 (-1.21–0.80) | 0.69 |
| 5 (16) | 110 | 3.66 | 131 | 3.83 | -0.03 (-0.75–0.70) | 0.94 |
| 6 (20) | 115 | 4.09 | 131 | 3.60 | -0.69 (-1.41–0.03) | 0.06 |

**Table 3. The impact of IPT with ASAQ on the breadth of malaria antigen recognition.**

| Study visit (Month) | N | Mean (Control) | N | Mean (ASAQ) | Difference in mean change in breadth of malaria antigen recognition from baseline (95% CI) | p value |
|---|---|---|---|---|---|---|
| 1 (0) (Baseline) | 60 | 2.27 | 61 | 1.95 | – | |
| 2 (4) | 115 | 3.42 | 112 | 3.28 | 0.18 (-0.57–0.92) | 0.65 |
| 3 (8) | 107 | 3.00 | 107 | 3.19 | 0.50 (-0.25–1.26) | 0.19 |
| 4 (12) | 29 | 2.52 | 33 | 2.58 | 0.37 (-0.67–1.42) | 0.48 |
| 5 (16) | 110 | 3.66 | 111 | 3.74 | 0.39 (-0.36–1.14) | 0.31 |
| 6 (20) | 115 | 4.09 | 111 | 4.03 | 0.26 (-0.49–1.01) | 0.50 |

## IgG reactivity to PfEMP1 domains (CIDRa1.1, CIDRa1.4 and CIDRa1.5)

The longitudinal analysis of IgG reactivity to the three domains of PfEMP1; CIDRa1.1, CIDRa1.4 and CIDRa1.5, showed no significant differences in change in reactivity between the DP and ASAQ groups compared to the control group during the first 16 months of the study (Fig 3D–3F and S1 Table). However, at the last visit (visit six) mean IgG reactivity to CIDRa1.4 and CIDRa1.5 in the DP group were significantly lower compared to the control group (mean OD -0.20 [95% CI -0.32 –-0.08] $p = 0.001$ and mean OD -0.10 [95% CI -0.20–0.00] $p = 0.05$, respectively) (Fig 3E and 3F).

## IgG reactivity in participants who attended all visits

Analysis of only schoolchildren who attended all visits, excluding visit four; n = 60, 52 and 49 in the DP, ASAQ and control groups, respectively, showed similar outcomes (S2 Fig and S2 Table). However, in the DP group, the change in mean IgG levels against MSP3 was significantly lower at visit five (mean OD -0.10 [95% CI -0.17 –-0.02] $p = 0.010$), and at visit six

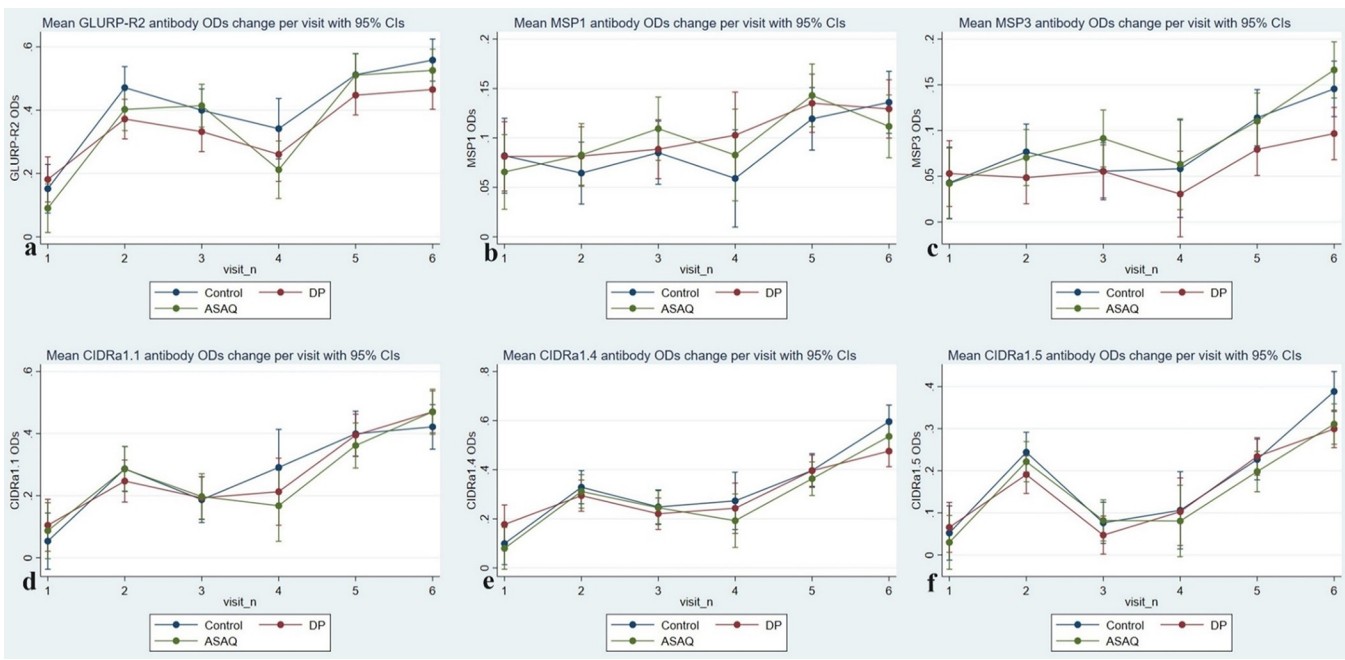

**Fig 3. Antibody trends against *P. falciparum* after IPTsc.** The study drugs, DP and ASAQ were given from visits 1 to 3 while during follow-up visits 4 to 6, no study drugs were given to all participants: (**a**) antibody trend against GLURP-R2, (**b**) antibody trend against MSP1, (**c**) antibody trend against MSP3, (**d, e, f**) antibody trend against CIDRa1.1, CIDRa1.4 and CIDRa1.5 of PfEMP1.

(mean OD -0.08 [95% CI -0.15 –-0.01] *p* = 0.033) while the differences in mean changes of IgG reactivity to CIDRa1.4 and CIDRa1.5 were not significant.

## Haemoglobin concentration, anaemia and malaria prevalence

The mean haemoglobin levels were significantly higher at visit four in the DP and ASAQ groups as compared to the control group and ((0.93 g/dL [95% CI 0.40–1.46] *p* = 0.001) and (0.98 g/dL [95% CI 0.44–1.54] *p* < 0.001), respectively) (S3A Fig). The prevalence of anaemia was lower in the DP and ASAQ groups at visit four compared with the control group, though not significant (-22.86 percentage points [95% CI -45.65–0.01] *p* = 0.06, and -18.67 percentage points [95% CI -40.50–3.16] *p* = 0.10, respectively). Lastly, the malaria prevalence was lower in both the DP and ASAQ group at visit four, but only significant in the DP group (-27.14 percentage points [95% CI -44.69–9.59] *p* = 0.003), ASAQ group (-19.13 percentage points [95% CI -39.6–0.79] *p* = 0.69), compared to the control group. At visit six, there was no difference in change of haemoglobin concentration, anaemia or malaria prevalence between the treatment and control groups as also seen in the larger trial study [10].

## Discussion

In a recently completed trial, IPTsc with DP or ASAQ was proven to be a feasible strategy to reduce malaria burden and related morbidities in schoolchildren in highly malaria-endemic areas of north-eastern Tanzania [10, 34]. The present study evaluated the IgG reactivity to a panel of six malarial antigens, including GLURP-R2, MSP1, MSP3, and PfEMP1 domains (CIDRa1.1, CIDRa1.4, CIDRa1.5), which are associated with both malaria exposure and protection [21, 22, 35, 36] during three 4-monthly interventions and up to one year after the last intervention to assess whether IPTsc with either DP or ASAQ interferes with the natural immune responses to these six antigens. The breadth of antibody response was shown to increase in both intervention groups during the study period, comparable to the control group, suggests that all schoolchildren are exposed to malaria parasites sufficiently enough to generate antibodies against the tested antigens. In particular, the schoolchildren in (ASAQ and DP) groups showed no significant difference in the breath of malaria antigen recognition from the control group across all study visits and no differences in the change of antibody levels for all individual antigens during and after the intervention period. Schoolchildren in the DP group, however, had a significantly lower antibody response to some of the tested antigens, and in particular, antibody responses to GLURP-R2 was significantly lower at all study visits except the fourth at month 12, when antibodies responses were non-significantly lower. In the DP group, antibody responses to MSP3, CIDRa1.4 and CIDRa1.5 were lower at month 20(visit 6) as well. The lower antibody reactivity and the significant reduction of malaria prevalence at visit four (the first visit after the intervention period) observed in the DP group is consistent with the effectiveness of the parasite clearance by the piperaquine component of the regimen owing to its long half-life [37]. These findings are consistent with those of the main trial study [10] as the reduction in prevalence of parasitaemia was significantly higher in the DP group compared to the ASAQ group at visit four. At visit six, there were no differences in the prevalence of parasitaemia between the groups [34], which indicates that the relatively lower levels of antibodies to GLURP-R2 and MSP3 in the DP group, one year after the intervention, were not associated with lower immunity to malaria.

To our knowledge, this is the first study to analyse the impact of IPTsc with DP or ASAQ on the antibody responses to the six tested antigens of *P. falciparum* and demonstrates that the development of immune responses to these *P. falciparum* antigens remains mostly unaffected by the interventions. These observations are consistent with findings from an IPTi with SP

study SP in Mozambique [16, 38] and a SMC study with SP plus AQ in Mali [39, 40]. These results suggest that IPTsc with DP or ASAQ is an effective strategy to reduce the malaria burden without interference with the development of IgG against malaria antigens in areas with high malaria incidence. Future studies may explore whether this holds true in low malaria-endemic areas as well. Additionally, findings from a related study based on this cohort suggest that the IPTsc strategy with DP or ASAQ does not appear to drive the selection of drug resistance markers specific to DP or ASAQ interventions [41]. However, the study observed an increase in *P. falciparum* multidrug resistance gene-1 (*Pfmdr1*) amplification and the prevalence of the 184F mutation which are likely attributable to the widespread use of artemether-lumefantrine as the first-line treatment for malaria in Tanzania [41, 42].

The current study has several limitations including not all participants from the main trial study were included, so the effect of IPTsc on the immune response of the entire cohort was not examined. Visit four, the first visit of the non-intervention phase, included two schools only due to the COVID-19 pandemic. Furthermore, this study only investigated six malaria antigens, whereas naturally acquired immunity to malaria involves responses to a broader range of antigens and different stages of the parasite life cycle [43, 44]. Although antibodies to the tested antigens are associated with protection from severe malaria [21, 45, 46], are important in controlling infections [23–26], and are targets for malaria vaccine development [47–50], the present study focus on a limited number of antigens may not fully capture the complexity of the immune response. The current study's strengths, includes rigorous data collection procedures and statistical analyses that accounted for missing data, and the use of validated assays, which together enhance the reliability and applicability of our findings.

## Conclusions

We have shown that effective IPTsc with either DP or ASAQ does not interfere with the development of antibodies to the different malarial antigens of the blood stages, expressed on the surface of infected erythrocytes and on merozoites and important targets of the naturally acquired immunity. Future studies may explore a wider range of malaria antigens using a much longer IPTsc programme.

## Supporting information

**S1 Fig. Participants included in the present study analysis.** From the previous study by Makenga et al. [10], 1566 schoolchildren were enrolled in randomized, controlled, open-label trial. The present study analysed samples from 369 schoolchildren, 119 participants were in the control group, 134 in the DP group and 116 in the ASAQ group.
(TIF)

**S2 Fig. Antibody trends against *P. falciparum* after intermittent preventive treatment for malaria in school-aged children (IPTsc).** Figures showing participants who attended all visits, and visit 4 has been excluded in the analysis: **(a)** antibodies trend against the glutamate rich protein-region 2 (GLURP-R2), **(b)** antibodies trend against the merozoite surface protein 1 (MSP1), **(c)** antibodies trend against the merozoite surface protein 3 (MSP3), **(d, e, f)** antibodies trend against three cysteine-rich interdomain region (CIDR) domains (CIDRa1.1, CIDRa1.4 and CIDRa1.5) of *P. falciparum* erythrocyte membrane protein 1 (PfEMP1).
(TIF)

**S3 Fig.** The impact of intermittent preventive treatment on malaria in school-aged children (IPTsc) on (a) change of mean haemoglobin, (b) anaemia prevalence and (c) malaria

prevalence from baseline, at each visit per study group (n = 369).
(TIF)

**S1 Table. Linear mixed effect analysis on the impact of IPTsc intervention on antibody response against six *P. falciparum* antigens in schoolchildren aged between 5–15 years, n = 369.**
(DOCX)

**S2 Table. Linear mixed effect analysis on the impact of IPTsc intervention on antibody response against six *P. falciparum* antigens in schoolchildren aged between 5–15 years.** Analysis based on participants who attended all visits, and visit 4 has been excluded in the analysis, n = 161.
(DOCX)

## Acknowledgments

The authors thank all study participants and staff who worked in the clinical trial. We thank staff from all organisations including the National Institute for Medical Research and the collaborating Ministries of the Government of Tanzania, who made this study possible. We thank Professor Michael Theisen (University of Copenhagen), for supplying the MSP1, MSP3, and GLURP-R2 domains.

## Author Contributions

**Conceptualization:** Eric Lyimo, Geofrey Makenga, Christian W. Wang.

**Data curation:** Eric Lyimo, Geofrey Makenga, Daniel T. R. Minja.

**Formal analysis:** Eric Lyimo, Geofrey Makenga, Christian W. Wang.

**Funding acquisition:** Geofrey Makenga, John P. A. Lusingu, Jean-Pierre Van geertruyden, Daniel T. R. Minja, Christian W. Wang, Vito Baraka.

**Investigation:** Eric Lyimo, Geofrey Makenga, John P. A. Lusingu, Jean-Pierre Van geertruyden, Daniel T. R. Minja, Christian W. Wang, Vito Baraka.

**Methodology:** Eric Lyimo, Louise Turner, Thomas Lavstsen, Christian W. Wang.

**Project administration:** Geofrey Makenga, John P. A. Lusingu, Jean-Pierre Van geertruyden, Daniel T. R. Minja, Vito Baraka.

**Resources:** Eric Lyimo, Geofrey Makenga, Louise Turner, Thomas Lavstsen, John P. A. Lusingu, Jean-Pierre Van geertruyden, Daniel T. R. Minja, Christian W. Wang, Vito Baraka.

**Supervision:** Geofrey Makenga, John P. A. Lusingu, Jean-Pierre Van geertruyden, Daniel T. R. Minja, Christian W. Wang, Vito Baraka.

**Validation:** Eric Lyimo, Vito Baraka.

**Visualization:** Eric Lyimo, Christian W. Wang.

**Writing – original draft:** Eric Lyimo.

**Writing – review & editing:** Geofrey Makenga, Louise Turner, Thomas Lavstsen, John P. A. Lusingu, Jean-Pierre Van geertruyden, Daniel T. R. Minja, Christian W. Wang, Vito Baraka.

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
