## [Decision Letter · Decision Letter 0]

28 Aug 2024

PONE-D-24-14848The impact of Intermittent Preventive Treatment among School aged Children with Dihydroartemisinin Piperaquine and Artesunate Amodiaquine on IgG response against six blood stage Plasmodium falciparum antigensPLOS ONE

Dear Dr. Lyimo,

Thank you for submitting your manuscript to PLOS ONE. After careful consideration, we feel that it has merit but does not fully meet PLOS ONE’s publication criteria as it currently stands. Therefore, we invite you to submit a revised version of the manuscript that addresses the points raised during the review process.

Line 93: Ghanaian

Line 101: … samples from 369 schoolchildren who were seen in at in more than one visit. This statement is not clear

Line 97: Describe clearly how the participants were recruited. It's very difficult to appreciate the procedure the way you have presented the information. Consider using a flow chat to make it clear.

Line 114: How were the schools selected?

Line 165: indicate how you obtained the Danish blood DBS

168: it is not enough to show the location of how the scoring was done. Indicate how it was done before showing readers where the original study was done

Methods section

Show clearly the inclusion/exclusion criteriaIndicate all confounding variables that are likely to affect the results reportedHow were lost to follow-up participants dealt withIndicate the details of the drugs used for this study (dihydroartemisinin-piperaquine and artesunate-amodiaquine). Brand name, source, dosage, and other pharmacokinetic details of the drugsThe authors were not clear on how the study participants were randomly selected for the studyUsing maps to show the location of the study sites/schools would be appropriateStudy design for this study is explicitly missing. You rather stated for the previous study which is different from this oneStudy was silent on how the malaria parasitemia was determinedAuthors did not show how anaemia was classified

Results

Table 1: define DP and ASAQ as table footnoteHemoglobin level: mean plus SD level will be more informativeTables 2/3: I find these tables a bit misleading. The control group from line 207 is 119. However, from these two tables, I see 60 participants. The same discrepancies were observed for DP/ASAQ.I also find the follow-ups very confusing. 60 participated at baseline, however, the subsequent visits were more. What does that mean? Did you recruit more after the baseline recruitment?What do the mean values represent?Bring some clarity into this.==============================

We look forward to receiving your revised manuscript.

Kind regards,

Enoch Aninagyei, PhD

Academic Editor

PLOS ONE

2. In the online submission form, you indicated that [The full datasets used and analysed in this study are available from the corresponding author (EL) on reasonable request and following the Ethics Committee and institutional data transfer policy.]. 

Additional Editor Comments (if provided):

Reviewers' comments:

Reviewer's Responses to Questions

**Comments to the Author**

1. Is the manuscript technically sound, and do the data support the conclusions?

Reviewer #1: Yes

Reviewer #2: Yes

Reviewer #3: Yes

2. Has the statistical analysis been performed appropriately and rigorously? 

Reviewer #1: Yes

Reviewer #2: Yes

Reviewer #3: Yes

3. Have the authors made all data underlying the findings in their manuscript fully available?

Reviewer #1: Yes

Reviewer #2: Yes

Reviewer #3: Yes

4. Is the manuscript presented in an intelligible fashion and written in standard English?

Reviewer #1: Yes

Reviewer #2: Yes

Reviewer #3: Yes

5. Review Comments to the Author

Reviewer #1: • The title is title informative and relevant

• The abstract is clear

• The references are recent

• In the methods:

Where are the doses of the drugs and their references?

How do you test the toxicity of the drugs?

All participants must be treated with albendazole at the baseline to account for the possible effect of helminths and schistosomiasis on anemia in the study area

I think IPT could P. falciparum-resistant strains.

The manuscript needs English editing

Why do you not use quantitative array technology(qSAT)to measure IgG1-4responses as the development of IgG1 and IgG3 cytophilic Abs against merozoite in infants or in children has been associated with the control of malaria infection(Adamou et al.,2019; Courtin et al 2009)

Reviewer #2: I think the paper is generally well written and should be considered for publication.

Below are some comments and questions:

1. Was the Human IgG ELISA performed in replicates? Please indicate in the Methods

2.Line 101...Was there special reason for using 369 participants? Did you perform any calculations to determine if this sample number is sufficient?

3.line 101 "... who were seen in at in more than one visit..."  "...who were seen at more than one visit..."

4. Line 70... Please define IPTc. Is IPTc = IPTsc?

Reviewer #3: General: The manuscript presents immunologic information on IgG responses to six blood stage Plasmodium falciparum antigens following Intermittent Preventive Treatment in school aged children (IPTsc) with Dihydroartemisinin Piperaquine (DP) and Artesunate Amodiaquine (ASAQ). The Title is apt while the Abstract clearly summarizes the study.

The manuscript describes a technically sound piece of research based on scientific concepts and acceptable methodology. The conclusions are based on and supported by the data presented. However, there are some aspects that need strengthening. Please see the uploaded reviewed manuscript with track changes (PONE-D-24-14848 [Rev.]) for detailed comments/suggestions/corrections on all sections of the manuscript, a few of which are given below.

English Language Quality: Above average but requires editing to expiate for the many typos and grammatical infelicities.

Study design, site and participants: The authors need to improve on the English language quality in describing the study design. There were ambiguities in the use of ”This study” under study design, site and participants (Please refer to Lines 100, 109) and elsewhere in the manuscript. Succeeding a foregoing sentence with ”This study” connotes reference to the earlier sentence. Better to use ”This present study” for their study to differentiate it from the earlier study trial.

Lines 59-60: Delete the words Intermittent preventive treatment of malaria in school-aged children as indicated here and subsequently, having been fully mentioned earlier in lines 53-54.

Lines 147/148: The first sentence should be deleted as indicated.

Lines 157-160: The highlighted sentence is vague and needs to be recast for better comprehension.

Lines 201-204: The sentence should be deleted as indicated.

Lines 204-205: The sentence should be recast/rephrased as indicated.

Lines 283: Change study group and the control groups to "treatment and control groups"

Lines 288-291: The highlighted phrase; “the exposure and protection of malaria” is confusing and should be clarified.

Lines 324-327: The sentence is incoherent and should be recast/rephrased for clarity.

Tables: Some revisions to the Tables have been indicated (Please see track changes in reviewed manuscript).

References: Okay. However, Plasmodium falciparum should be italicized throughout as highlighted in the reviewed manuscript.

6. PLOS authors have the option to publish the peer review history of their article (what does this mean?). If published, this will include your full peer review and any attached files.

Reviewer #1: No

Reviewer #2: **Yes: **Dr. Daniel Addo-Gyan

Reviewer #3: No

---

## [Author Response · Author response to Decision Letter 0]

20 Sep 2024

Response to Editor and Reviewers have been included in the Response to Reviewer rebuttal letter.

---

## [Decision Letter · Decision Letter 1]

6 Nov 2024

PONE-D-24-14848R1The impact of Intermittent Preventive Treatment in School aged Children with Dihydroartemisinin Piperaquine and Artesunate Amodiaquine on IgG response against six blood stage Plasmodium falciparum antigensPLOS ONE

Dear Dr. Lyimo,

Thank you for submitting your manuscript to PLOS ONE. After careful consideration, we feel that it has merit but does not fully meet PLOS ONE’s publication criteria as it currently stands. Therefore, we invite you to submit a revised version of the manuscript that addresses the points raised during the review process.

We look forward to receiving your revised manuscript.

Kind regards,

Enoch Aninagyei, PhD

Academic Editor

PLOS ONE

Journal Requirements:

Reviewers' comments:

Reviewer's Responses to Questions

**Comments to the Author**

1. If the authors have adequately addressed your comments raised in a previous round of review and you feel that this manuscript is now acceptable for publication, you may indicate that here to bypass the “Comments to the Author” section, enter your conflict of interest statement in the “Confidential to Editor” section, and submit your "Accept" recommendation.

Reviewer #4: (No Response)

2. Is the manuscript technically sound, and do the data support the conclusions?

Reviewer #4: Yes

3. Has the statistical analysis been performed appropriately and rigorously? 

Reviewer #4: Yes

4. Have the authors made all data underlying the findings in their manuscript fully available?

Reviewer #4: Yes

5. Is the manuscript presented in an intelligible fashion and written in standard English?

Reviewer #4: Yes

6. Review Comments to the Author

Reviewer #4: All comments are included in the manuscript. If authors address the comments successfully, it should be accepted for publication.

7. PLOS authors have the option to publish the peer review history of their article (what does this mean?). If published, this will include your full peer review and any attached files.

Reviewer #4: **Yes: **Tanko Rufai

---

## [Author Response · Author response to Decision Letter 1]

14 Nov 2024

Dear Editor,

Regarding our submitted manuscript (ID: PONE-D-24-14848) titled “The impact of Intermittent Preventive Treatment among in aged Children with Dihydroartemisinin Piperaquine and Artesunate Amodiaquine on IgG response against six blood stage Plasmodium falciparum antigens,” we are grateful for the valuable comments and suggestions provided by you and the four reviewers. We have carefully considered the constructive feedback and incorporated the necessary revisions into our manuscript. We are pleased to submit the revised version along with this rebuttal letter.

Please find below our point-to-point responses to all issues that were raised by the reviewers. Appropriate changes have also been incorporated in the revised manuscript. 

Thank you for the recommendation regarding the deposition of laboratory protocols in protocols.io. After reviewing our manuscript, we believe that the detailed overview of our procedures already provides sufficient clarity and reproducibility for future researchers. Given that the methods are thoroughly described, we feel that an additional protocol submission may not be necessary in this case.

Again, thank you for your note regarding the data sharing requirements. While we fully support the principles of data transparency, the dataset underlying this study contains sensitive information that falls under the jurisdiction of our Ethics Committee and institutional data transfer policies. Making this data publicly available would potentially compromise the conditions of the protocol approved by our research ethics board.

We therefore request an exemption from the requirement to publicly deposit our dataset. We are happy to make the data available to other researchers upon reasonable request, in accordance with the institutional and ethical guidelines governing research involving human participants, especially minors. Further, data can be requested, through our ethics committee, and we have revised the text that will be included in the manuscript regarding the data sharing the passage read as follows; The full datasets used and analysed in this study are available upon reasonable request from the corresponding author (EL), subject to approval by the Medical Research Coordinating Committee (MRCC) of the National Institute for Medical Research (NIMR) in Tanzania. In accordance with the Ethics Committee’s guidelines and institutional data transfer policy, the MRCC mandates that all data collected within Tanzania cannot be transferred or shared without their authorization and the signing of a data transfer agreement. Researchers who meet the criteria for data access may request the datasets by contacting: ethics@nimr.or.tz. 

We confirm that all changes and revisions were reviewed and approved by all authors. We look forward to your continued support.

Yours sincerely,

Eric Lyimo

Editor’s comments

Line 93: Ghanaian

Response: We thank the editor for noticing the error and the correct word has been inserted, line 101.

Line 101: … samples from 369 schoolchildren who were seen in at in more than one visit. This statement is not clear.

Response: The sentence has been revised and it is now clear, line 110.

Line 97: Describe clearly how the participants were recruited. It's very difficult to appreciate the procedure the way you have presented the information. Consider using a flow chat to make it clear.

Response: A flowchart has been added as a supporting information as (S1 Sig), and referenced, line 111.

Line 114: How were the schools selected?

Response: We thank the Editor for the comment on school selection, the information has been added, line 127 – 129.

Line 165: indicate how you obtained the Danish blood DBS

Response: The has been added, line 213 – 216.

168: it is not enough to show the location of how the scoring was done. Indicate how it was done before showing readers where the original study was done.

Response: We thank the reviewer for the comments, the information on the scoring has been added, line 220 – 227.

Methods section

1. Show clearly the inclusion/exclusion criteria

Response: The information on inclusion and exclusion criteria have been added in brief, line 10, further on how samples were excluded from the present study analysis was already on line 270 – 275.

2. Indicate all confounding variables that are likely to affect the results reported

Response: We are thankful for the comment and Information on the study limitations have been mentioned, line, 392 – 398 and 402 – 410.

3. How were lost to follow-up participants dealt with

Response: The present study included the per-protocol analysis from the primary trial study, which includes only participants who fully adhered to the study protocol, without deviations or missing data. However, we had missing samples or samples that were excluded from laboratory analyses, the missing data were taken care by the robustness of the linear mixed model as outline on the statistical section line 240 – 242.

4. Indicate the details of the drugs used for this study (dihydroartemisinin-piperaquine and artesunate-amodiaquine). Brand name, source, dosage, and other pharmacokinetic details of the drugs

Response: The details on the drugs has been added, line 154 – 169.

5. The authors were not clear on how the study participants were randomly selected for the study

Response: The information on participant selection has been added as supporting information, S1 Fig, and referenced line 111.

6. Using maps to show the location of the study sites/schools would be appropriate

Response: We are thankful for the comment, a map for the study sites has been submitted with the revised manuscript, figure description line 124.

7. Study design for this study is explicitly missing. You rather stated for the previous study which is different from this one

Response: We thank the editor for the comment, and the information on study design has been added line 105 - 106.

8. Study was silent on how the malaria parasitemia was determined

Response: The information on how parasitaemia was determined has been added, line 187 – 190. Further, the procedure for assessing parasitaemia was detailed in primary study procedures; the clinical study protocol (https://doi.org/10.1016/j.conctc.2020.100546), which has been cited in the current manuscript line 190, and a briefly explained line 187 – 190. 

9. Authors did not show how anaemia was classified

Response: We are thankful for the comment, and the information on how anaemia was determined has been added, line 181 – 184.

Results

1. Table 1: define DP and ASAQ as table footnote

Response: DP and ASAQ have been defined as table footnote, line 277.

a. Hemoglobin level: mean plus SD level will be more informative

Response: Haemoglobin levels values have been changed to mean and SD from median and IQR values, table1.

2. Tables 2/3: I find these tables a bit misleading. The control group from line 207 is 119. However, from these two tables, I see 60 participants. The same discrepancies were observed for DP/ASAQ.

a. I also find the follow-ups very confusing. 60 participated at baseline, however, the subsequent visits were more. What does that mean? Did you recruit more after the baseline recruitment?

Response: We are thankful for this comment. Regarding the concerns about the discrepancy in participant numbers in Tables 2 and 3, we would like to clarify the following:

The control group indeed consisted of a total of 119 participants across all visits. However, due to factors such as quality control (QC) failures for some samples and the availability of samples at each visit, the number of participants whose data were analysed varies between visits. A participant may have attended a visit, but their samples might not appear in the final analysis if they did not meet the QC standards or if the sample was unavailable for analysis.

Additionally, the use of a linear mixed model in our analysis allows us to account for any missing data, ensuring that the statistical power and validity of the model are preserved despite these fluctuations in sample numbers.

b. What do the mean values represent?

Response: The mean values in the tables represent the average outcome of antibody levels at each visit for the participants included in the analysis, after accounting for any missing data and variability between individuals. In our case, control vs. treatment groups, the mean values provide a comparison between the groups across visits, adjusting for any baseline differences that may influence the results. 

c. Bring some clarity into this.

Response: The variations in participant numbers across visits are due to sample quality control (QC) failures and sample availability as explained above (2a), which led to some samples being excluded from the analysis. However, the linear mixed-effects model (LMM) we used is designed to handle missing data efficiently, meaning that it still provides reliable estimates of the mean outcome for each visit. The model adjusts for individual differences (through random effects) and ensures that the missing data does not bias the results. Therefore, the mean values in the tables represent the average outcomes for each visit, considering all available data, and they remain valid even when the number of participants analysed varies between visits.

Response to Reviewers

Reviewer #1

The title is title informative and relevant

Response: We thank the reviewer for the comment.

• The abstract is clear

Response: We thank the reviewer for the comment.

• The references are recent

We thank the reviewer for the comment.

• In the methods:

Where are the doses of the drugs and their references?

Response: We thank the reviewer for the comment, and the information has been added, line 161 – 176.

How do you test the toxicity of the drugs?

Response: The present study did not specifically focus on testing the toxicity of study drugs. However, both drugs are WHO prequalified, meaning they have undergone rigorous testing for safety, quality, efficacy and have been subjected to drug trials. Additionally, these drugs are already on the market. In the primary study, the drugs were administered according to the manufacturer’s guidelines, with healthcare professionals and trained schoolteachers overseeing the process. Participants were also monitored for any adverse reactions under the supervision of study nurses to ensure their safety during treatment.

All participants must be treated with albendazole at the baseline to account for the possible effect of helminths and schistosomiasis on anaemia in the study area.

Response: We thank the reviewer for the comment and possible effect of helminths and schistosomiasis on anaemia was accounted and drugs were administered, line 171 – 176.

I think IPT could P. falciparum-resistant strains.

We thank the reviewer for the comment, but it is not clear, however we think it should have been "I think IPT could [lead to] P. falciparum-resistant strains,". Therefore, to address the issue drug resistance, a separate study was conducted (https://doi.org/10.1016/j.ijid.2024.107102) on this cohort to assess the potential selection of drug resistance markers and found no significant selection pressures. The results were published after submission of this article, but we have added texts to explain this concern, and the publication is now cited, line 392 – 398.

The manuscript needs English editing

Response: We thank the reviewer for the comment and the English language has been improved throughout the manuscript.

Why do you not use quantitative array technology(qSAT)to measure IgG1-4responses as the development of IgG1 and IgG3 cytophilic Abs against merozoite in infants or in children has been associated with the control of malaria infection(Adamou et al.,2019; Courtin et al 2009)

Response: We thank the reviewer for the comment and suggestions, but the technology was not available in the current study setting; however we have measured the total reactive IgG.

Reviewer #2:

I think the paper is generally well written and should be considered for publication.

Response: We are thankful for the comment from the reviewer.

Below are some comments and questions:

1. Was the Human IgG ELISA performed in replicates? Please indicate in the Methods

Response: We thank the reviewer for the comment, the IgG ELISA was not performed in replicates, each of the samples done as a single measurement, line 204 – 205.

2.Line 101...Was there special reason for using 369 participants? Did you perform any calculations to determine if this sample number is sufficient?

Response: We thank the reviewer for the comment. This study was an exploratory study that used the maximum number of samples available. The new figure, S1 Fig show the how we obtain the 369 participants, referenced line 111.

3.line 101 "... who were seen in at in more than one visit..."  "...who were seen at more than one visit..."

Response: The sentence has been corrected, line 110.

4. Line 70... Please define IPTc. Is IPTc = IPTsc?

Response: We thank the reviewer for the comment, IPTc is intermittent preventive treatment in children (specifically preschool children), and IPTsc as defined cited throughout the manuscript is the intermittent preventive treatment in schoolchildren. IPTc which also referred as SMC, starts to appear in line 55.

Reviewer #3

General: The manuscript presents immunologic information on IgG responses to six blood stage Plasmodium falciparum antigens following Intermittent Preventive Treatment in school aged children (IPTsc) with Dihydroartemisinin Piperaquine (DP) and Artesunate Amodiaquine (ASAQ). The Title is apt while the Abstract clearly summarizes the study.

The manuscript describes a technically sound piece of research based on scientific concepts and acceptable methodology. The conclusions are based on and supported by the data presented. However, there are some aspects that need strengthening. Please see the uploaded reviewed manuscript with track changes (PONE-D-24-14848 [Rev.]) for detailed comments/suggestions/corrections on all sections of the manuscript, a few of which are given below.

Response: We thank the reviewer for the remarks.

English Language Quality: Above average but requires editing to expiate for the many typos and grammatical infelicities.

Response: We thank the reviewer for the comments, the language has been improved throughout the manuscript. 

Study design, site and participants: The authors need to improve on the English language quality in describing the study design. There were ambiguities in the use of ”This study” under study design, site and participants (Please refer to Lines 100, 109) and elsewhere in the manuscript. Succeeding a foregoing sentence with ”This study” connotes reference to the earlier sentence. Better to use ”This present study” for their study to differentiate it from the earlier study trial.

Response: We thank the reviewer of the comment and “This study” has been replaced with “The present study” throughout the manuscript.

Lines 59-60: Delete the words Intermittent preventive treatment of malaria in school-aged children as indicated here and subsequently, having been fully mentioned earlier in lines 53-54. 

Response: We thank the reviewer for the comment, this was actually supposed to be IPTc, which is intermittent preventive treatment in children (studies have investigated SMC in preschool children/under five years old and referred to it as IPTc), line 55. Therefore, IPTsc appears for the first time in line 62 – 63.

Lines 147/148: The first sentence should be deleted as indicated.

Response: The sentence has been deleted as requested, line 192.

Lines 157-160: The highlighted sentence is vague and needs to be recast for better comprehension.

Response: We thank the reviewer for the comments, the sentence has been amended, line 202 – 205.

Lines 201-204: The sentence should be deleted as indicated.

Response: We thank the reviewer for the suggestion, the sentence has been deleted, line 263 – 266.

Lines 204-205: The sentence should be recast/rephrased as indicated.

Response: We thank the reviewer for the suggestion, the sentence has been rephrased, line 26

---

## [Editor Report · Decision Letter 2]

12 Dec 2024

The impact of Intermittent Preventive Treatment in School aged Children with Dihydroartemisinin Piperaquine and Artesunate Amodiaquine on IgG response against six blood stage Plasmodium falciparum antigens

PONE-D-24-14848R2

Dear Dr. Eric Lyimo,

We’re pleased to inform you that your manuscript has been judged scientifically suitable for publication and will be formally accepted for publication once it meets all outstanding technical requirements.

Kind regards,

Enoch Aninagyei, PhD

Academic Editor

PLOS ONE
---

## [Editor Report · Acceptance letter]

11 Jan 2025

PONE-D-24-14848R2 

PLOS ONE

Dear Dr. Lyimo, 

I'm pleased to inform you that your manuscript has been deemed suitable for publication in PLOS ONE. Congratulations! Your manuscript is now being handed over to our production team.

Kind regards, 

on behalf of

Dr Enoch Aninagyei 

Academic Editor

PLOS ONE